# Systematic Comparison of Plant Promoters in *Nicotiana* spp. Expression Systems

**DOI:** 10.3390/ijms232315441

**Published:** 2022-12-06

**Authors:** Ekaterina S. Shakhova, Nadezhda M. Markina, Tatiana Mitiouchkina, Evgenia N. Bugaeva, Tatiana A. Karataeva, Kseniia A. Palkina, Liliia I. Fakhranurova, Ilia V. Yampolsky, Karen S. Sarkisyan, Alexander S. Mishin

**Affiliations:** 1Shemyakin-Ovchinnikov Institute of Bioorganic Chemistry, Russian Academy of Sciences, 117997 Moscow, Russia; 2Department of Translational Medicine, Pirogov Russian National Research Medical University, 117997 Moscow, Russia; 3Synthetic Biology Group, MRC London Institute of Medical Sciences, London W12 0NN, UK; 4Institute of Clinical Sciences, Faculty of Medicine and Imperial College Centre for Synthetic Biology, Imperial College London, London W12 0NN, UK

**Keywords:** EGFP, BY-2, plant cell packs, transcription unit, transient assay

## Abstract

We report a systematic comparison of 19 plant promoters and 20 promoter-terminator combinations in two expression systems: agroinfiltration in *Nicotiana benthamiana* leaves, and *Nicotiana tabacum* BY-2 plant cell packs. The set of promoters tested comprised those not present in previously published work, including several computationally predicted synthetic promoters validated here for the first time. The expression of EGFP driven by different promoters varied by more than two orders of magnitude and was largely consistent between two tested Nicotiana systems. We confirmed previous reports of significant modulation of expression by terminators, as well as synergistic effects of promoters and terminators. Additionally, we observed non-linear effects of gene dosage on expression level. The dataset presented here can inform the design of genetic constructs for plant engineering and transient expression assays.

## 1. Introduction

In engineering projects, iteration time is the critical parameter that affects development timelines. Plant engineering is among the slowest areas of biotechnology, with iteration times starting with months and reaching years for some plant species. This aspect of plant engineering puts additional pressure on predictability of performance of regulatory elements used in genetic designs, yet the molecular tools used in plant research remain less characterized than those used in microbial and animal biotechnology.

To date, a number of promoters for expression in plant systems have been described. Promoter sequences of viral origin are among the most commonly used [1], but they are thought to be more prone to gene silencing and may suffer from uneven expression in plant tissues [2]. A number of endogenous plant promoters have also been characterized [3], although regulatory elements of plant origin generally provide lower expression levels and may be prone to tissue-specific regulation. More recently, several synthetic promoter designs have been reported [4,5,6], with expression levels reaching those of viral promoters and showing more even expression profiles across plant tissues. 

Systematic comparisons of some of the viral and endogenous plant promoters are available [7,8,9], providing a valuable starting point for design and refactoring of genetic constructs. Among those, Tian and colleagues [9] reported a comparison of 105 combinations of 45 promoters and 13 terminators in both *N. benthamiana* leaves and BY-2 cell culture, using dual luciferase reporter assay. This study revealed a 326-fold difference in the performance of tested combinations, demonstrating high correlation between the levels of mRNA and luminescence output. The study focused on promoters of agrobacterial, plant, and viral origins, leaving synthetic promoters out of the comparison. 

Previous studies also highlighted different ability of terminators to contribute to expression efficiency, with mechanisms including post-transcriptional gene silencing, which manifested even in transient expression assays [10]. Significant non-independence of promoter-terminator combinations has been reported for *Arabidopsis thaliana* MM1 cell culture [8], highlighting the importance of terminator choice in plant engineering. 

In this work, we aimed to (1) supplement available data with our systematic comparison of 19 plant promoters, most of which were not present in previous comparative studies, and (2) assay effects of promoter-terminator combinations. As prototyping for plant engineering is typically done in transient expression systems, we also aimed to (3) assess the coherence of the two *Nicotiana*-based model transient expression systems: upon agroinfiltration of *N. benthamiana* leaves, and in recently introduced high-throughput assay in *N. tabacum* BY-2 plant cell packs [11].

## 2. Results

### 2.1. Comparison of Promoters

We selected a set of 19 promoters: viral—p35s_0.4kb, pFMV, pCmYLCV, synthetic and previously validated *in planta*—pMinSyn104, pMinSyn105, pMinSyn108, pMinSyn110, predicted synthetic—pMinSyn159x, pMinSyn1556x, pMinSyn1569x, pMinSyn1637x, pMinSyn1824x, pMinSyn1904x, and native plant promoters—pAtTCTP, pdel5_MtHP, pAtUBQ10, pAtRPS5a, pAtPD7, pAtAct2 [5,7,12,13,14,15,16,17,18] (Table 1). 

To compare the strength of the promoters, we assembled transcription units with the following structure: promoter–[5′UTR]–EGFP–tOCS (Appendix A) and assayed green fluorescence as a proxy for EGFP expression level. For plant promoter sequences which were published without clear annotation of the transcription start, in order not to risk misidentifying the end of the promoter, entire published sequences, including native 5′UTR, were taken into comparison. For promoters lacking endogenous 5′UTR sequences, we included 5′UTR from *Arabidopsis thaliana* RBCS2B gene downstream of the promoter sequence (Table 1), with the only exception of pAct2, which contained tobacco mosaic virus 5′UTR and was obtained directly from the Plant Parts kit (Addgene, Watertown, MA, USA, #1000000047). 

We performed a comparison of EGFP expression in *N. benthamiana* leaves and BY-2-based plant cell packs in the background of expression of RNA silencing suppressor p19 [19]. The fluorescence levels span about two orders of magnitude in both systems, with viral and synthetic promoters generally providing significantly higher expression levels than plant endogenous promoters (Figure 1). 

Fluorescence levels conferred by MinSyn synthetic promoters were high, exceeding those of endogenous plant promoters, with MinSyn108 and MinSyn110 being the strongest promoters of the set and providing expression similar to p35S promoter. The low autofluorescence background of the cell packs allowed for gene dosage experiments in a wider range of expression levels. Starting from commonly used OD600 = 0.5, we lowered the OD of the Agrobacterium suspension down to just OD600 = 0.005 (Figure 2). For some promoters the fluorescence scaled linearly in a log-log scale, while for other ones we observed deviations from the expected trend (Figure 2, inset).

The value of the Spearman’s rank correlation coefficient (R_s_ [20] = 0.72, *p* = 0.00027) computed for the entire dataset indicates strong correlation of the expression levels in both systems, with pAtAct2 being the most prominent outlier (Figure 3). In both systems, the 0.4 kb version of the promoter p35S outperformed the other promoters.

### 2.2. Comparison of Terminators

We then aimed to test whether promoters and terminators affect expression levels non-independently. For four selected promoters with varied expression levels—pAtTCTP, pAtUBQ10, pMinSyn159x, and pdel5_MtHP—we assembled plasmids with various terminators: tRBCS3C, tHSP18.2, tAtAct2, and tATPase. In the case of the relatively weak promoter pAtTCTP, expression level modulation by terminators was less pronounced compared to stronger promoters. For one of the strongest promoters, pdel5_MtHP, the change in terminator resulted in more than a 50-fold change in expression level, highlighting the importance of choosing 3′UTR and terminator. Among the tested terminators, tOCS, tHSP18.2, and tATPase provided high expression overall, with tOCS performing better across two systems. In contrast, tAtAct2 and tRBCS3C resulted in low expression in both systems (Figure 4).

## 3. Discussion

We compared multiple viral, plant, and synthetic promoters in *N. benthamiana* leaves and *N. tabacum* BY-2 cell packs. Typically, in plant biotechnology, a high level of expression is achieved by using viral promoters, most commonly, promoter p35S. However, some strong plant promoters, such as pAtUBQ10 and pdel5_MtHP, showed strength comparable with viral promoters pFMV and pCmYLCV, but these promoters were still weaker than p35S. Other plant promoters—AtTCTP and AtPD7—did not demonstrate such high expression levels.

As expected from previous reports, synthetic promoters of the MinSyn family generally demonstrated high levels of expression. We showed that several synthetic promoters, which were not experimentally tested before, were functional in *Nicotiana* spp. expression systems. The strength of some of the synthetic promoters approached that of p35S. The small size, high strength, and relative tissue-independence of expression of MinSyn promoters set them aside as the new top choice for many types of plant engineering work.

We also performed a small-scale comparison of terminator efficiencies in the context of several promoters, adding to the discussion of synergistic effects of promoter and terminator on expression [8]. We identified tOCS as an overall winner among the tested terminators, in contrast to the previous reports highlighting tHSP18.2 as an optimal terminator in *A. thaliana* and *O. sativa* [18].

Finally, in this work, we performed experiments in two model systems used for prototyping of genetic designs in plant cells. Our results suggest that the commonly used *N. benthamiana* agroinfiltration assay can be functionally replaced with the more high-throughput and less laborious screening in BY-2-based plant cell packs. In addition, lower signal variation in plant cell packs allows for higher assay resolution.

## 4. Materials and Methods

### 4.1. Plasmids

Some promoters and terminators were obtained from Addgene kits: MoClo Plant Parts Kit (Kit #1000000047), MoClo Plant Parts II and Infrastructure Kit (Kit #1000000135). All backbone plasmids were obtained from MoClo Toolkit (Kit #1000000044). Synthetic fragments of MinSyn, AtTCTP, del5_MtHP, FMV, CmYLCV promoters, and HSP18.2 terminator were ordered synthetically from Twist Bioscience (USA) and cloned into pICH4123 (MinSyn promoters), pICH4129 (AtTCTP, del5_MtHP, FMV, CmYLCV promoters), or pICH41276 (HSP18.2 terminator), using Golden Gate troubleshooting protocol described in [20].

Promoter AtPD7 was assembled by annealing primers (pr748 + pr749) and (pr750 + pr751) (Appendix A) and then cloned into pAGM1311, and then shuttled into the final vector pICH41295.

The sequences of all plasmids used in this study are available in Appendix A.

### 4.2. Transformation of Agrobacterium tumefaciens

Level 1 plasmids were transformed into *Agrobacterium tumefaciens* strain AGL0 and grown on LB agar plates containing 50 mg/L of rifampicin and 200 mg/L of carbenicillin. Individual colonies were then inoculated into 10 mL of LB medium containing the same quantity of antibiotics. After overnight incubation at 28 °C with 220 rpm, shaking cultures were centrifuged at 2900× *g* and resuspended in 25% glycerol and stored as glycerol stocks at −80 °C.

### 4.3. Nicotiana benthamiana Plants

Potted *N. benthamiana* plants were grown in a controlled environment indoor facility at 22 ± 2 °C under long day conditions (16 h light/8 h dark photoperiod, 120–150 μmol s^−1^ m^−2^) and 50–60% relative humidity.

### 4.4. Nicotiana tabacum BY-2 Cell Culture

BY-2 cell culture was grown in BY-2 medium (MS (Merck, Saint Loius, MO, USA, M5524) with 0.2 mg/L 2.4D (Duchefa, Haarlem, Netherlands, D0911), 200 mg/L KH_2_PO_4_ (KupavnaReactiv, Staraya Kupavna, Russia, similar to Merck 60220-M), 1 mg/L thiamine (Duchefa, Haarlem, The Netherlands, T0614), 100 mg/L myo-inositol (Merck, Saint Loius, MO, USA, I7508), and 30 g/L sucrose (PanReac AppliChem, Darmstadt, Germany, 141621)) at 27 °C by shaking at 130 rpm in darkness, with 2 mL of 1-week old culture being transferred in new 200 mL of BY-2 medium every week [21].

### 4.5. Transient Transformation of Nicotiana benthamiana Leaves

Before agroinfiltration, glycerol stocks of agrobacteria were inoculated into 10 mL of LB containing 50 mg/L of rifampicin and 200 mg/L of carbenicillin, as well as 100 μM of acetosyringone (Merck, Saint Loius, MO, USA, D134406). The cultures were grown in the dark overnight at 28 °C with 220 rpm shaking. The cultures were then centrifuged at 2900 g, suspended in MMA buffer (10 mM MES, Formedium, Norfolk, England, MES03; 10 mM MgCl_2_, Molecula, Krasnodar, Russia, 29218779; 200 μM acetosyringone Merck, Saint Loius, MO, USA, D134406), and incubated at 28 °C, 100 rpm for 3–4 h. Next, optical density at 600 nm was measured and used to dilute each culture to the optical density of 0.6. In addition, suspension of agrobacteria containing a plasmid encoding pNOS–P19–tOCS was added at the optical density of 0.2. The final optical density at 600 nm of agrobacterial suspension used for infiltration was 0.8. We then used these cultures to infiltrate leaves of 4–6-week-old *N. benthamiana*, using a 1 mL medical syringe without needle. At least four leaves of at least three different plants were infiltrated for each experiment. The exact numbers of leaves are included in the figure’s legends.

### 4.6. Imaging of N. benthamiana Leaves

Seventy-two hours after agroinfiltration, *N. benthamiana* leaves were detached and the fluorescence was measured from the bottom side of each leaf in the ChemiDoc MP Imaging System (BioRad). The following settings were used for imaging: “blue epi illumination” excitation light and green emission captured with a 530/28 nm filter. We then used ImageJ to analyze images [22].

### 4.7. Transient Transformation of BY-2 Cell Culture

Transformations of BY-2 cells were made following the protocol from [23]. One-week-old BY-2 culture was pelleted in black 96-well plates to create cell packs and infiltrated by a mixture of two agrobacterial strains: one expressing pNOS–P19–tOCS (OD600 = 0.2) in binary vector and another one expressing GFP under control of different promoters and terminators (OD_600_ = 0.5). To obtain OD_600_ = 0.05 and 0.005 of agrobacteria with EGFP, this mixture was serially diluted. Plates were incubated at 80% humidity at 22 °C for 72 h before fluorescence measurements. Six (Figure 1) or four (Figure 2) wells containing BY-2 cell packs were infiltrated per construct.

### 4.8. Measuring EGFP Fluorescence in BY-2 Cells

Seventy-two hours after agrobacterial infection, 96-well plates containing BY-2-based cell packs were placed into a microplate reader (Tecan Spark, Männedorf, Switzerland), and fluorescence was measured with 469/20 nm excitation and 514/20 nm emission settings. We used a gain of 40, 30 flashes, and integration time of 40 µs to detect fluorescence.

### 4.9. Data Presentation and Statistics

Data are plotted as box-and-whiskers plots implemented in Seaborn (https://pypi.org/project/seaborn/0.12.1/ accessed on 19 October 2022) package (version 0.11.2). The boxes extend from the lower to upper quartile values of the data, and the line represents the median, whiskers represent the full data range (Figure 1, Figure 2 and Figure 4). Alternatively, mean values with whiskers covering standard deviation (Figure 3) were plotted. The Scikit-post-hoc Python package (https://pypi.org/project/scikit-posthocs/0.7.0/ accessed on 10 May 2022, version 0.7.0) was used for Kruskal–Wallis tests followed by multiple pairwise post-hoc Conover’s tests, with *p* values corrected by the step-down method using Sidak adjustments. Sample numbers (N) are reported in the figure legend.

## 5. Conclusions

The reported dataset contributes to the available data on the characterization of genetic elements for plant engineering. We performed cross-comparison of each transcription unit in two *Nicotiana*-based systems often used to prototype genetic design for the generation of stable genetically modified plants. The lower variance and higher throughput of plant cell pack assay in a multiwell format may eventually lead to the replacement of the de facto standard leaf agroinfiltration as a method of choice for transient assay. The systematic comparison of the transcription units of various designs in leaf infiltration and plant cell packs assays reported here supports this trend. We believe that the dataset presented here will be useful for the design of highly active constitutively expressed transcription units.

## Figures and Tables

**Figure 1 ijms-23-15441-f001:**
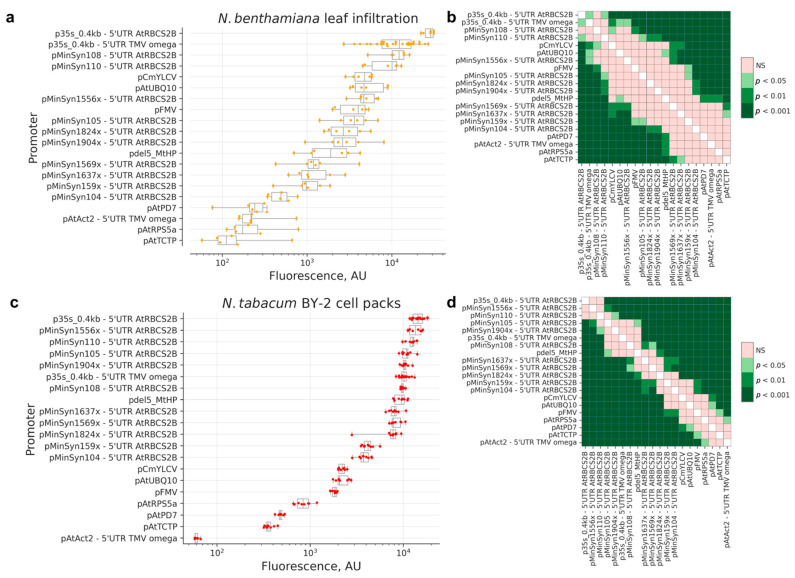
Comparison of expression of EGFP, driven by various promoters. The tOCS terminator was used in all transcription units shown on the graph. (**a**,**b**) Agroinfiltration in *N. benthamiana* leaves; *N* = 6 leaves for each promoter, except p35s_0.4kb—TMV omega (*N* = 30 leaves) (**c**,**d**) Transient transformation of *N. tabacum* BY-2-based plant cell packs. *N* = 6 cell packs for each promoter; (**a**,**c**)—box and whiskers plots, the box extends from the lower to upper quartile values of the data, the vertical line represents the median. Whiskers represent a full data range; (**b**,**d**) color-coded *p*-values of Conover’s test. NS—non-significant.

**Figure 2 ijms-23-15441-f002:**
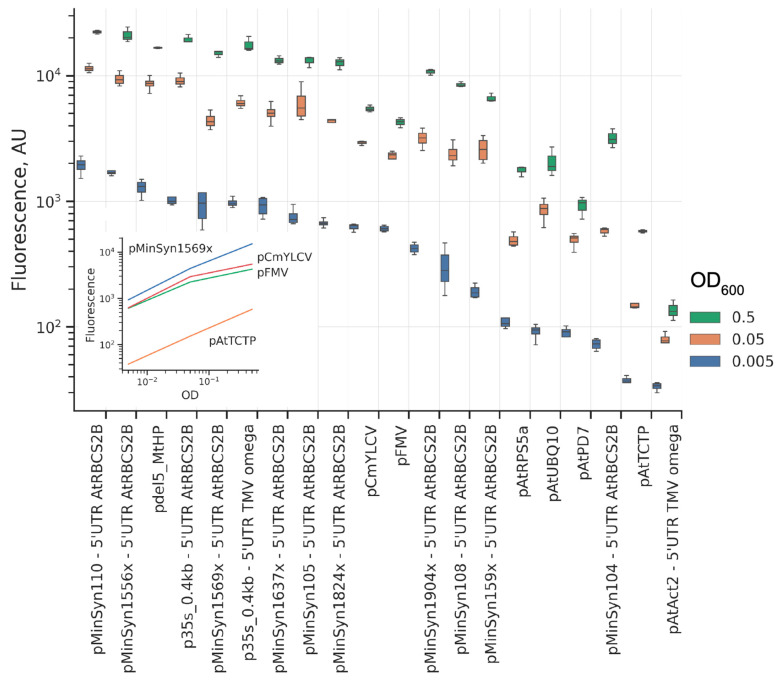
Effect of the gene dosage on the EGFP expression in BY-2 cell packs. The colors of the boxes correspond to the OD_600_ of EGFP-carrying Agrobacterium upon inoculation: 0.5 (green), 0.005 (brown), and 0.0005 (blue). The order of promoters is based on level of fluorescence with OD_600_ = 0.005 (from strong to weak); (inset) Representative log-log plots showing linear (pMinSyn1569x, pAtTCTP) and non-linear (pCmYLCV, pFMV) changes in fluorescence upon the increase in gene dosage. *N* = 4 cell packs were measured for each data point.

**Figure 3 ijms-23-15441-f003:**
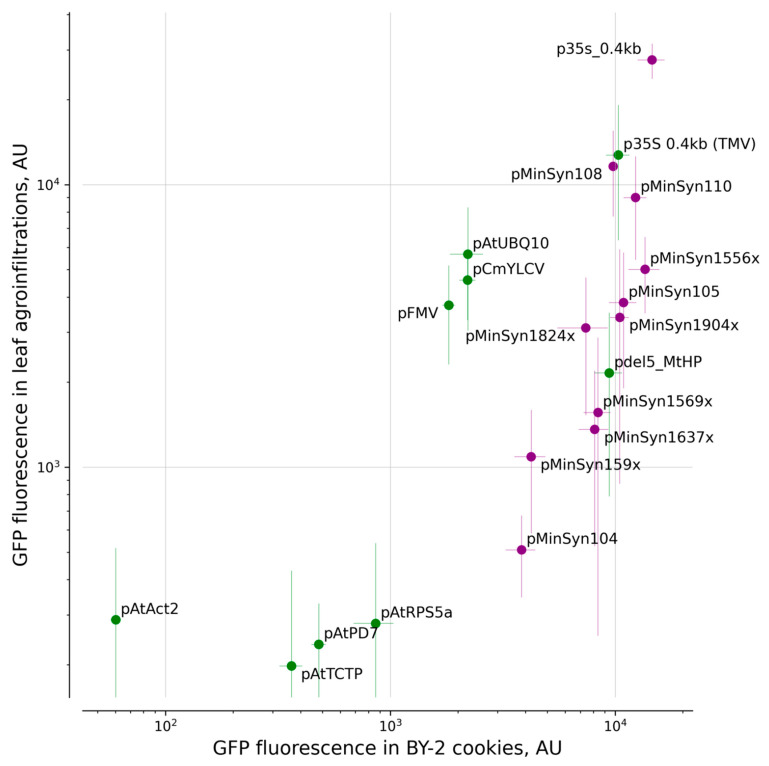
Comparison of EGFP expression within various transcription units between BY-2 plant cell packs and *N. benthamiana* transient expression systems. All transcription units consisted of EGFP under the control of designated promoters and either AtRBCS2B 5′UTR (synthetic promoters and p35S 0.4 kb, in purple) or native 5′UTR sequences (and TMV omega 5′UTR for p35S 0.4 kb, green dots). The tOCS terminator was used in all transcription units shown on the graph. The same dataset as Figure 1 was used. Points indicate mean values; whiskers show standard deviation.

**Figure 4 ijms-23-15441-f004:**
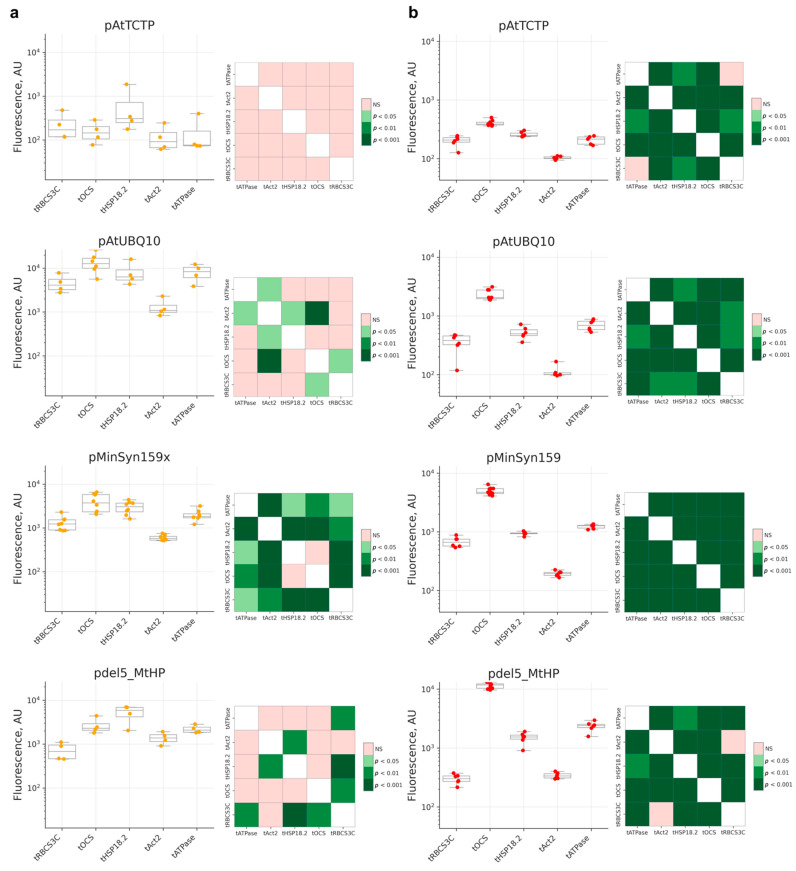
Comparison of terminators in the context of different promoters in two expression systems: (**a**) *N. benthamiana* leaf agroinfiltration (*N* = 4 leaves for each promoter) and (**b**) *N. tabacum* BY-2 plant cell packs (*N* = 6 cell packs for each promoter). Box and whisker plots (**left**) on each panel are accompanied by color-coded *p*-values of Conover’s test (**right**). NS—non-significant. The box extends from the lower to upper quartile values of the data, with the horizontal line representing the median. Whiskers represent a full data range.

**Table 1 ijms-23-15441-t001:** Promoters and terminators used in this study.

Element	Name	Description	Reference
Promoter	p35s_0.4kb–5′UTR TMV omega	Promoter 35s (Cauliflower Mosaic Virus), 0.4 kb in combination with 5′UTR, omega (Tobacco Mosaic Virus) or 5′UTR, RbcS2B (AT5g38420, *A. thaliana*)	[7]
	p35s_0.4kb–5′UTR AtRBCS2B		
	pFMV	Figwort mosaic virus (FMV) 34S promoter + 5′UTR	[12]
	pCmYLCV	Cestrum yellow leaf curling virus (CmYLCV) promoter + 5′UTR	[13]
	pMinSyn104–5′UTR AtRBCS2B	Minimal synthetic promoters + 5′UTR, RbcS2B (AT5g38420, *A. thaliana*)	[5]
	pMinSyn105–5′UTR AtRBCS2B		
	pMinSyn108–5′UTR AtRBCS2B		
	pMinSyn110–5′UTR AtRBCS2B		
	pMinSyn159x–5′UTR AtRBCS2B	Predicted minimal synthetic promoters + 5′UTR, RbcS2B (AT5g38420, *A. thaliana*)	[5] *
	pMinSyn1556x–5′UTR AtRBCS2B		
	pMinSyn1569x–5′UTR AtRBCS2B		
	pMinSyn1637x–5′UTR AtRBCS2B		
	pMinSyn1824x–5′UTR AtRBCS2B		
	pMinSyn1904x–5′UTR AtRBCS2B		
	pAtTCTP	Small constitutive promoter from Arabidopsis translationally controlled tumor protein (AtTCTP) gene (0.3 kb) + 5′UTR	[14]
	pdel5_MtHP	Constitutive promoter upstream of MtHP gene isolated from *Medicago truncatula* + 5′UTR **	[15]
	pAtUBQ10	800 bp fragment upstream of Arabidopsis UBQ10 (At4g05320)—strong, constitutive promoter + 5′UTR	[16]
	pAtRPS5a	1700 bp upstream of Arabidopsis RPS5a (At3g11940)—strong, constitutive promoter + 5′UTR	[16]
	pAtPD7	Strong and constitutive promoter from the Arabidopsis serine carboxypeptidase-like gene AtSCPL30 (456 bp upstream of the translation initiation codon ATG of AtSCPL30) + 5′UTR	[17]
	pAtAct2–5′UTR TMV omega	Promoter AtAct2 (AT3G18780, *A. thaliana*) + 5′UTR, omega (Tobacco Mosaic Virus)	[7]
Terminator	tOCS	3′UTR + terminator OCS (*A. tumefaciens*)	[7]
	tAtAct2	3′UTR + terminator Act2 (*A. thaliana*)	
	tRBCS3C	3′UTR + terminator RBCS3C (*S. lycopersicum*)	
	tATPase	3′UTR + terminator ATPase (*S. lycopersicum*)	
	tHSP18.2	3′UTR + the heat shock protein 18.2 (HSP) terminator of *Arabidopsis thaliana*	[18]

* Promoters from the preprint (https://doi.org/10.1101/2020.05.14.095406, accessed on 15 May 2020) are denoted with “x” (i.e., pMinSyn159x) to avoid confusion with the ones reported in the final version of Ref. [5]. ** Sequence of this promoter differs by 4 nt, see Appendix A.

## Data Availability

We will make new plasmids used in this manuscript available on Addgene.

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
