# Peer review of "Systematic Comparison of Plant Promoters in Nicotiana spp. Expression Systems"

_ijms, 2022, doi:10.3390/ijms232315441_

Round 1

Reviewer 1 Report

This study aims to improve the expression assays in plant systems by identifying additional promoter and promoter-terminator combinations in 2 expression systems namely N. benthamiana and N. tabacum BY-2 plant cell packs. Here, the authors performed a systematic comparison of 19 plant promoters and 20 promoter-terminator combinations using a dual luciferase reporter system. This study demonstrates that some synthetic promoters which have not been characterised before could be added to the repertoire of promoters to be used in plant engineering experiments.

Comments:

The strength of this study is in the systematic comparison approach to characterise promoters and promoter-terminator combinations especially the synthetic promoters that have been previously identified bioinformatically but have not been tested in planta. The authors claim that some of the synthetic promoters approached the level that of p35S. The authors stated that some pMinSyn showed significant higher expression levels than endogenous promoters however, no statistical analysis was done to show specifically which ones are higher. This applies to both N. benthamiana and N. tabacum BY-2 plant cell packs.  In Figure 1a, pMinSyn1569x, pMinSyn1637x, pMinSyn159x seem to have the same effect as pATct2-TMV. Statistical analysis will help determine if the values are not overlapping and therefore significant. Also, I am wondering why the authors used different 5’-UTRs (AtRB, TMV and endogenous), why not use the same 5’UTR since AtRB seem to have better expression: p35s-5’ UTR AtRB vs p35s-5’ UTR TMV omega in Fig 1a: again stats analysis will determine if the values are same or not. The authors did not state in the results nor discussion section why they have opted to use different 5’-UTRs. If the same UTR is used in all promoters tested, then there will be no confusion on the effect of the 5’UTRs in fluorescence. The pMinSyn should be compared to the other promoters with the same 5’UTR.

The authors indicated that promoters were selected because they have been reported to have high expression levels. This is confusing since most pMinSyn promoters have not been tested before. Have the pMinSyn been assayed before for expression in vitro? Of the predicted promters, only 1 out 6 (pMinSyn1556x) seem to have similar expression as p35S.

For the promoter-terminator combination, again statistical analysis is needed. Also, Why was pMinSyn108 and pMinSyn1556x not included in the promoter-terminator study? This promoters might benefit from having a compatible terminator to have higher expression than p35S. As such this synthetic promoters would be a great alternative to p35S, which was also not included in this experiment. 

Lastly, this paper aims at offering alternative promoters that would be capable of driving high expression, but all of the promoters tested are the same level, at best, as the commonly used p35S promoter. Thus, pMinSyn doesn’t seem to offer anything new.

Other comments:

1.      N. benthamiana and N. tabacum are not italicised in Figure 1 a and b, and also in the figure legend. N. benthamiana not italicised in Results section following Figure 1.

2.      In the Materials and Methods, section 4.1 Plasmids: Synthetic promoters were obtained synthetically, indicate if they were purchased and indicate what company

3.      Section 4.8 Measuring EGFP fluorescence in BY-2 cells: I would assume that Tecan Spark is a brand of microplate reader; “ ..place in a microplate reader (Tecan Spark).

Author Response

"The authors stated that some pMinSyn showed significant higher expression levels than endogenous promoters however, no statistical analysis was done to show specifically which ones are higher. This applies to both N. benthamiana and N. tabacum BY-2 plant cell packs.  In Figure 1a, pMinSyn1569x, pMinSyn1637x, pMinSyn159x seem to have the same effect as pATct2-TMV. Statistical analysis will help determine if the values are not overlapping and therefore significant."
RESPONSE: We thank the reviewer for bringing our attention to the lack of statistical analysis of significance of difference between the promoters. We now performed the analysis and reflected it on the Figures and in the Methods sections. Furthermore, in answering the question of significance of differences  we abandoned the unnecessary data normalisation steps and improved data presentation in figures 1, 3, 4. Figures 1 and 4 now include heatmaps allowing for visual interpretation of P-values in pairwise comparisons between different designs of  transcriptional units.

"Also, I am wondering why the authors used different 5’-UTRs (AtRB, TMV and endogenous), why not use the same 5’UTR since AtRB seem to have better expression: p35s-5’ UTR AtRB vs p35s-5’ UTR TMV omega in Fig 1a: again stats analysis will determine if the values are same or not. The authors did not state in the results nor discussion section why they have opted to use different 5’-UTRs. If the same UTR is used in all promoters tested, then there will be no confusion on the effect of the 5’UTRs in fluorescence. The pMinSyn should be compared to the other promoters with the same 5’UTR."
RESPONSE: We agree with the reviewer that the consistent use of the same 5’UTR would allow for more insights into comparison of transcriptional strength of the promoters, compared to what we can observe for promoter-5’UTR combinations. 
We aimed to keep the same 5’UTR where possible, but for native “promoters” sequences which were published without clear annotation of transcription start, for practical reasons, we decided to not risk guessing the end of the promoter, taking entire published sequences into comparison. Stacking two 5’UTRs – a native one and our standard 5’UTR – did not make sense to us either, as the joint effect of stacked 5’UTR sequences would not make the results easily interpretable. Thus, we decided to proceed with endogenous 5’UTR. We now added clarification of that choice into the manuscript

"The authors indicated that promoters were selected because they have been reported to have high expression levels. This is confusing since most pMinSyn promoters have not been tested before. Have the pMinSyn been assayed before for expression in vitro? Of the predicted promters, only 1 out 6 (pMinSyn1556x) seem to have similar expression as p35S."
RESPONSE: We agree that our wording was misleading, as some of the pMinSyn promoters were indeed not tested before. Specifically, promoters pMinSyn104, pMinSyn105, pMinSyn108 and pMinSyn110 were tested in vitro before, and promoters pMinSyn159x, pMinSyn1556x, pMinSyn1569x, pMinSyn1637x, pMinSyn1824x and pMinSyn1904x were not. We now changed the corresponding paragraph to address this comment. 

"For the promoter-terminator combination, again statistical analysis is needed. Also, Why was pMinSyn108 and pMinSyn1556x not included in the promoter-terminator study? This promoters might benefit from having a compatible terminator to have higher expression than p35S. As such this synthetic promoters would be a great alternative to p35S, which was also not included in this experiment." 
RESPONSE: Statistical analysis is now included in the study of the promoter-terminator combinations. pMinSyn108 and pMinSyn1556x were left out of these experiments to keep the amount of genetic contructs manageable given our resources, and to be able to focus on confirming previous studies. 

"Lastly, this paper aims at offering alternative promoters that would be capable of driving high expression, but all of the promoters tested are the same level, at best, as the commonly used p35S promoter. Thus, pMinSyn doesn’t seem to offer anything new."
RESPONSE:  This is not correct, the aim of this study was not to find a promoter driving stronger expression than the 35S.  In this work, we aimed to compare multiple promoters in the same experimental setting, showing among other findings that none of these promoters were stronger than the 35S in our systematic comparison.  We amended the text in the potentially misleading paragraph 2.1 

"Other comments:

  1.     N. benthamiana and N. tabacum are not italicised in Figure 1 a and b, and also in the figure legend. N. benthamiana not italicised in Results section following Figure 1."
    RESPONSE: Now corrected.
  1.      "In the Materials and Methods, section 4.1 Plasmids: Synthetic promoters were obtained synthetically, indicate if they were purchased and indicate what company"
    RESPONSE: Now corrected.
  1.      "Section 4.8 Measuring EGFP fluorescence in BY-2 cells: I would assume that Tecan Spark is a brand of microplate reader; “ ..place in a microplate reader (Tecan Spark)."
    RESPONSE: Now corrected.

Reviewer 2 Report

To,

The Editor,

IJMS, MDPI,

Manuscript ID: ijms-2046006

 Subject: Submission of comments of the manuscript in “IJMS"

 Dear Editor IJMS, MDPI,

 Thank you very much for the invitation to consider a potential reviewer for the manuscript (ID: ijms-2046006). My comments responses are furnished below as per each reviewer’s comments. 

In the reviewed manuscript, authors reported a systematic comparison of 19 plant promoters and 20 promoter-terminator combinations in two expression systems: agroinfiltration in N. benthamiana leaves, and N. tabacum BY-2 plant cell packs. The set of promoters tested comprised those not present in previously published work, including several computationally predicted synthetic promoters validated here for the first time. Expression of EGFP driven by different promoters varied by more than two orders of magnitude and was largely consistent between two tested Nicotiana systems. We confirmed previous reports of significant modulation of expression by terminators, as well as synergistic effects of promoters and terminators. Additionally, we observed non-linear effects of gene dosage on expression level. The dataset presented here can inform the design of genetic constructs for plant engineering and transient expression assays. In general, the manuscript represents a very big piece of information. Therefore, it might be conditionally accepted subject to minor revision. Authors have to improve their manuscripts with many non-clear meanings, inaccuracies and inconsistencies, and the authors need to address the following issues before it can be accepted for publication. 

1.    General note: the figures in this section are quite low resolution and difficult to make out. Higher-resolution versions will be needed for publication, for example, in Figures 1, 4, and 6.

2.    Discussion- many times references are made to the information given in the Introduction section (sometimes more general information). It would be good to discuss especially the results and critically, ie. Which can cause differences in the results of authors and other articles.

3.    I would like the Authors to provide the methodology and results of the number of replications wherever possible. The same applies to the statistical significance of the results. Please describe statistical methods used in the work in materials and methods.

4.    Authors must add the conclusion.

5.    References: shall have to correct the whole References according to the ”Instructions for the Authors”, e.g. title should not be in italics, the Journal name is in italics, and the author shall have to use the abbreviated name Journals cited the year must be bold, the scientific name must be italics etc. Please check all references carefully.

6.    Reference 1 Current Plant Biology was replaced with Curr. Plant Biol. Hence, please check all references carefully.

Author Response

"Therefore, it might be conditionally accepted subject to minor revision. Authors have to improve their manuscripts with many non-clear meanings, inaccuracies and inconsistencies, and the authors need to address the following issues before it can be accepted for publication. 

  1.   General note: the figures in this section are quite low resolution and difficult to make out. Higher-resolution versions will be needed for publication, for example, in Figures 1, 4, and 6."

RESPONSE: We believe the low resolution of the figures may have resulted from the atomatic pdf conversion upon submission. We double checked all the figures and now provide them at the resolution of at least 300 dpi.

  1.    "Discussion- many times references are made to the information given in the Introduction section (sometimes more general information). It would be good to discuss especially the results and critically, ie. Which can cause differences in the results of authors and other articles."

RESPONSE: We extended the discussion, as suggested by the reviewer.

  1.    "I would like the Authors to provide the methodology and results of the number of replications wherever possible. The same applies to the statistical significance of the results. Please describe statistical methods used in the work in materials and methods."

RESPONSE: We thank the reviewer for bringing our attention to the lack of necessary details and the need for statistical analysis. We now updated the Methods section to provide the missing information about the number of replicates, as well as information about statistical methods used. We have also redesigned Figures 1 and 4 to include visual clues for significance of pairwise comparisons (P-values as heatmaps).

  1.    "Authors must add the conclusion."

RESPONSE: We thank the reviewer for this suggestion. We added the conclusion section to the manuscript. 

  1.    "References: shall have to correct the whole References according to the ”Instructions for the Authors”, e.g. title should not be in italics, the Journal name is in italics, and the author shall have to use the abbreviated name Journals cited the year must be bold, the scientific name must be italics etc. Please check all references carefully."

RESPONSE: Now corrected. 

  1.   Reference 1 Current Plant Biology was replaced with Curr. Plant Biol. Hence, please check all references carefully.

RESPONSE: Now corrected. 

Round 2

Reviewer 1 Report

The authors have addressed all the questions. I recommend the publication of the manuscript in its current form (except for a very minor comment:  "Arabidopsis thaliana" in the results section, paragraph 2  should be italicised).